# Effects of 1,25-Dihydroxycholecalciferol Glycoside Supplementation on the Growth, Intestinal Health, and Immunity of Broilers from Breeders Supplemented or Not with the Same Additive

**DOI:** 10.3390/vetsci12050434

**Published:** 2025-05-01

**Authors:** Thiago S. Andrade, Nilton Rohloff Junior, Paulo L. O. Carvalho, Bruno S. Vieira, José G. Vargas Junior, Arele A. Calderano, Paulo C. Pozza, Leandro D. Castilha, Elcio S. Klosowski, Cinthia Eyng, Ricardo V. Nunes

**Affiliations:** 1Department of Animal Science, Western Parana State University (Unioeste), Marechal Candido Rondon 85960-000, PR, Brazil; nilton_rohloff_8@hotmail.com (N.R.J.); paulo.levioc@unioeste.br (P.L.O.C.); elcio.klosowski@unioeste.br (E.S.K.); cinthiaeyng@unioeste.br (C.E.); nunesrv@hotmail.com (R.V.N.); 2Department of Animal Science, Federal University of Uberlandia, Uberlandia 38408-100, MG, Brazil; vieirabs@ufu.br; 3Department of Animal Science, Federal University of Espírito Santo, Alegre 29500-00, ES, Brazil; jose.vargas@ufes.br; 4Department of Animal Science, Universidade Federal de Viçosa, Viçosa 36570-900, MG, Brazil; calderano@ufv.br; 5Department of Animal Science, State University of Maringá, Maringá 87020-900, PR, Brazil; pcpozza@uem.br (P.C.P.); ldcastilha@uem.br (L.D.C.)

**Keywords:** gene expression, nutrition, poultry, vitamin D_3_

## Abstract

The objective of this study was to evaluate the effects of 1,25-Dihydroxycholecalciferol glycoside supplementation on the growth, intestinal health, and immunity of broilers. The results showed that supplementation of breeders led to heavier chicks at hatch, better feed conversion, and improved nutrient absorption in the intestines. There was also an improvement in immune indicators, suggesting that this supplementation may contribute to healthier and more efficient growth of broilers.

## 1. Introduction

In modern broilers, rapid muscle growth combined with still-developing skeletal support can lead to a higher incidence of metabolic challenges, bone issues, and locomotor difficulties throughout their growth [1,2,3]. At the same time, the pursuit of significant weight gain in birds can result in biomechanical imbalances in skeletal development, leading to problems such as restricted mobility, lameness, and other bone deformities [4,5]. Notably, tibial dyschondroplasia is observed during the early growth phase, while structural issues and distortions in long bones may occur in the later stages [6,7,8]. These complexities in the development of modern broiler chickens require special attention to minimizing potential negative impacts on their health and well-being during the production cycle [9,10].

In this context, vitamin supplementation plays a crucial role in reducing mobility issues and bone deformities [11,12], particularly vitamin D_3_, which acts as an essential metabolic cofactor, significantly influencing bone metabolism and the regulation of calcium and phosphorus in birds [4,13,14]. Studies suggest that vitamin D_3_ supplementation in diets can reduce locomotion problems, minimize skeletal disorders, and contribute to improved growth and immunity in poultry [15,16,17].

Among vitamin D_3_ metabolites, 1,25-Dihydroxycholecalciferol, the active form of vitamin D_3_, plays a critical role in calcium and phosphorus absorption in the intestinal tract of birds [1,4,9,15,18,19,20], promoting efficient mineral uptake in the body [17,21,22]. This function is essential for proper skeletal development, ensuring bone formation and health in birds [23,24,25].

1,25-Dihydroxycholecalciferol can also be found in glycosylated form in certain plants, such as *Solanum glaucophyllum* [1,9,10,15,26]. In this glycosylated form (1,25(OH)_2_D_3_-G), the vitamin D_3_ compound is bound to glucose molecules [27], giving it a prolonged and targeted action in the birds’ intestines [7,10,27,28,29]. When consumed, birds can metabolize this glycosylated form of 1,25-Dihydroxycholecalciferol, allowing for the gradual release of the active substance, extending its activity time in the birds’ bodies and improving its efficacy in regulating calcium and phosphorus metabolism [15,30]. This herbal form offers a natural alternative for vitamin D_3_ supplementation, with specific properties that can benefit the digestive system and absorption of these essential minerals for growth [15,31,32].

Recent research highlights that 1,25(OH)_2_D_3_-G supplementation not only significantly improves bone mineralization in birds, promoting a stronger and more resilient skeletal structure [9,32,33,34,35], but also plays a key role in enhancing the efficient transport of essential minerals such as calcium and phosphorus. Additional studies emphasize the positive influence of 1,25(OH)_2_D_3_-G in facilitating the absorption of these minerals in the birds’ intestines, contributing to bone health and proper skeletal development [13,36,37,38]. These findings highlight the benefits of this form of supplementation, extending beyond bone mineralization to optimize mineral metabolism in birds. Moreover, 1,25(OH)_2_D_3_-G plays a vital role in strengthening the immune response, improving defense against pathogens [8,39,40].

This substance triggers a series of biochemical events that positively influence the body’s natural defenses [18,24]. Recent studies by researchers such as [10] highlight that adequate supplementation of 1,25(OH)_2_D_3_-G is associated with significant improvements in birds’ defense capabilities against pathogens. By enhancing the immune response, 1,25(OH)_2_D_3_-G can increase the effectiveness of natural barriers, such as mucosal integrity and the production of antimicrobial substances [5]. Furthermore, this active substance can modulate the gene expression of cytokines and other immune mediators, contributing to a more balanced and responsive immune environment [19,20,36].

In this context, the importance of 1,25(OH)_2_D_3_-G in the immune response of birds extends beyond its traditional role in calcium homeostasis and bone phosphorylation [9]. Its positive influence on pathogen defense stands out as a crucial factor in maintaining overall poultry health, providing significant benefits for poultry production [18,39,41,42,43]. However, determining the optimal dosage for incorporation into bird diets remains a point of contention [5,19,20,44]. The lack of precise definition in this area presents a substantial challenge, as inadequate doses may result in toxicity issues [4,6,39,45,46]. This uncertainty underscores the importance of conducting studies to mitigate any toxicity risks and optimize the potential benefits offered by this form of vitamin D_3_ [15,20,21,32,47,48,49,50,51].

Therefore, this study hypothesizes that breeder supplementation with 1,25(OH)_2_D_3_-G enhances bone mineralization in newly hatched broilers, promoting balanced growth and a strengthened immune system. To test this hypothesis, the study evaluated the effects of 1,25(OH)_2_D_3_-G supplementation on the performance, biochemical blood, bone health, intestinal histomorphometry, and gene expression in broilers from breeders supplemented or not with this additive.

## 2. Materials and Methods

### 2.1. Animal Ethics Statement

The procedures were approved by the Ethics Committee for the Use of Animals of the National Council for the Control of Animal Experimentation—UNIOESTE (Protocol No. 01/2021) and were previously approved by the National Council for the Control of Animal Experimentation in accordance with Normative No. 37 of 15 February 2018.

### 2.2. Animals and Experimental Design

Two broiler breeder houses (G1 and G2) were established in the municipality of Pato Branco, Paraná, Brazil, with the aim of housing 8000 (Ross 308 AP—AP95, Avícola Pato Branco, PR, Brazil) broiler breeders at 21 weeks of age. Each house measured 175 m × 13 m, was constructed with a metallic structure, was covered with Brasilit roofing, and had a ceiling height of 3 m without insulation. The houses were equipped with bird-proof mesh and white curtains, along with fans to maintain positive pressure ventilation. The nests were mechanical, while the feeding system consisted of automatic chain feeders specifically designed for breeders, and the drinkers were pendular models from the brand Plasson.

The diets provided to the birds in houses G1 and G2 were identical, formulated with corn and soybean meal to meet the nutritional requirements throughout the production period (21 to 62 weeks of age). However, the breeders in house G1 received a mash diet supplemented with 1,25(OH)_2_D_3_-G at a dosage of 100 mg/kg starting at 21 weeks of age, while the breeders in house G2 received a mash diet without 1,25(OH)_2_D_3_-G supplementation. This level was chosen based on its expected influence on calcium and phosphorus metabolism, aiming to enhance mineral transfer from breeders to embryos and improve skeletal health in the progeny (broilers).

At 62 weeks of age, the eggs from each barn were collected separately, identified, selected, and sanitized. The eggs were then transported to the hatchery and incubated in incubators (Jamesway Platinum Series, Jamesway Incubator Company Inc., Cambridge, ON, Canada), where they remained for a period of 18 days, maintained at a temperature of 37.5 °C and a relative humidity of 60%. After the 18-day incubation period, the eggs were transferred to the hatchers, where they remained at a temperature of 37.2 °C and a relative humidity of 65% until hatching occurred.

After hatching, the broiler chicks were sexed and vaccinated on their first day of life to prevent common poultry diseases, including Marek’s disease, Avian Pox, Gumboro disease, and Infectious Bronchitis. The vaccines administered included Vaxxitek HVT+IBD (Boehringer), which was applied subcutaneously for protection against Marek’s disease (HVT) and Gumboro. The Prevvaxion RN (Boehringer) vaccine, also administered subcutaneously, provided protection against Marek’s disease (Rispens). For fowl pox protection, the Bouba das Aves Suave (Biovet) vaccine was applied subcutaneously. Additionally, the Bronquite H120 (MSD Animal Health, Kenilworth, NJ, USA) vaccine, aimed at protecting against Infectious Bronchitis (Massachusetts strain), was administered via spray. All vaccines were applied according to the manufacturers’ recommended procedures, ensuring effective early immunization and protection of the broiler chicks against these diseases.

After vaccination, the broiler chicks were properly identified based on the breeder’s origin, considering whether or not they were supplemented with 1,25(OH)_2_D_3_-G. They were then sent to the Poultry Research Center at the State University of Western Paraná, in Marechal Cândido Rondon, Paraná, Brazil.

A total of 1152 one-day-old male, Ross 308 AP95 broiler chicks were distributed in a completely randomized design, arranged in a 2 × 3 factorial scheme. One of the factors was the origin of the broiler chicks from breeders fed with or without 1,25(OH)_2_D_3_-G (0 and 100 mg/kg), and the broiler chicks were fed with three inclusion levels (0, 50, and 100 mg/kg) of 1,25(OH)_2_D_3_-G) up to 21 days of age. This resulted in 6 treatments, with 8 replicates and 24 birds per experimental unit.

### 2.3. Diets and Feeding Management

The experimental diets (Table 1) were isonutritive and isocaloric, based on corn and soybean meal, following the nutritional recommendations of [52]. The vitamin used in this study (1,25-Dihydroxycholecalciferol glycoside) is from the product Panbonis^®^ 10, which is produced from *Solanum glaucophyllum* in powder form and standardized with pre-gelatinized wheat starch and wheat middlings. Its analytical composition includes 1,25-Dihydroxycholecalciferol glycosides (minimum of 10 ppm), moisture (maximum of 14%), crude protein (14–18%), crude fiber (5.25–8.75%), crude fat (3–6%), crude ash (3–6%), sodium (0.0–0.7%), lysine (0.6–0.9%), and methionine (0.2–0.4%). The product is manufactured by Herbonis Animal Health GmbH, located in Switzerland, and is marketed in Brazil through local distributors.

The feed provided to the birds throughout the experimental period was in meal form. The micronutrients, along with 1,25(OH)_2_D_3_-G, were weighed and manually mixed in 5 kg bags. The mixture was then transferred to the Premiata PRM vertical mixer, with a capacity of 500 kg, where the blending process with the other feed ingredients was carried out for 10 min to ensure homogeneous distribution. After mixing, the feed was bagged in 30 kg bags. The use of 1,25(OH)_2_D_3_-G replaced the inert material in the feed (g g^−1^). The 1,25(OH)_2_D_3_-G was provided only up to 21 days of age (Table 2). After this period, the broilers received the same grower and finisher diets without 1,25(OH)_2_D_3_-G. During the experimental period (1 to 21 days), the birds had *ad libitum* access to water and feed.

### 2.4. Performance

The birds and feed were weighed at the beginning of the experimental period and at 21 days of age to determine average feed intake, weight gain, and feed conversion ratio. Mortality was recorded daily to adjust feed intake and feed conversion ratio [53].

### 2.5. Intestinal Histomorphometry

At 21 days of age, one bird per experimental unit was euthanized by cervical dislocation for the removal of the digestive tract and subsequent evaluation of intestinal histomorphometry (villus height, crypt depth, villus-height-to-crypt depth ratio, and absorption area). The small intestine was exposed, and the jejunum was isolated for sampling. The segment used was the distal portion from the duodenal loop to Meckel’s diverticulum. A 2 cm fragment of the jejunum was collected 5 cm before Meckel’s diverticulum. This fragment was fixed in 10% buffered formalin, dehydrated in a series of increasing ethanol concentrations, and then embedded in paraffin. Semi-serial sections of 5 µm from each segment were placed on glass slides and stained using the hematoxylin-eosin technique [54].

The measurements were performed using the PROPLUS IMAGE 4.1 imaging system. On each slide, the length and width of the villi, as well as the depth and width of the crypts, were recorded. These morphometric measurements were used to calculate the absorption surface area of the intestinal mucosa [55]. The villus-height-to-crypt depth ratio was calculated using the results of villus height and crypt depth measurements.

### 2.6. Biochemical Variables

At 21 days of age, 4 mL of blood were collected from two birds per experimental unit for the analysis of calcium, phosphorus, lactate dehydrogenase (LDH), creatine phosphokinase (CPK), and alkaline phosphatase. Blood collection was performed by puncture of the ulnar vein while the birds were positioned in lateral recumbency. Specific vacuum tubes with a clot activator and a 5 mL capacity (CRAL, Cotia, SP, Brazil) were used, along with adapters and 25 × 0.8 mm needles (Labor Import, Weihai, Shandong, China).

After collection, the samples were kept in a horizontal position for 15 min and then centrifuged (K14-4000, Kasvi, São José dos Pinhais, PR, Brazil) at 2500 rpm (1050 g) for 10 min at room temperature. After serum separation, the samples were properly identified and transferred to 2 mL microtubes (CRAL, Cotia, SP, Brazil) and stored in a freezer at −20 °C until the time of analysis [56].

For the readings, the samples were thawed under refrigeration at 4 °C and kept in the refrigerator for a maximum of 8 h. Prior to analysis, the samples were centrifuged in a microcentrifuge (Eppendorf^®^, Minispin^®^ model, Hamburg, Germany) at 1050 g for 10 min at room temperature to remove any remaining fibrin traces. The biochemical variables were measured using an automatic biochemical analyzer with spectrophotometry, specifically the Flexor EL200 (Elitech^®^ brand, Flexor EL200 model, Puteaux, France). The analysis was conducted using reagents, calibrators (Elical II), and control standards (Elitrol I) from the Elitech^®^ brand.

The spectrophotometric methods demonstrated high sensitivity and precision for all analyzed analytes. The intra-assay coefficients of variation ranged from 1.1% to 3.6%, while the inter-assay values varied between 1.7% and 6.5%, ensuring reliability in the analyses. The correlation between methods was consistent, with r ≥ 0.998. Specifically, for calcium and phosphorus, intra-assay coefficients ranged from 1.1% to 2.8%, while inter-assay values were 1.7% to 3.7%. Lactate dehydrogenase (LDH) showed a sensitivity of 0.087 m∆A/min per U/L and high reproducibility, with intra-assay variations from 1.2% to 3.6% and inter-assay variations from 2.8% to 6.5%. Creatine phosphokinase (CK) and alkaline phosphatase (ALP) had intra-assay coefficients ranging from 1.1% to 3.5% and inter-assay values from 1.7% to 5.2%, reinforcing the accuracy of these methodologies.

### 2.7. Bone Quality and Tibia Proximal Composition

The tibias of euthanized birds (21 days old) were cleaned of flesh. The right tibia was weighed and measured with a digital caliper (Mitutoyo Absolute 0.01 mm–150 mm). Using the weight and length of the bone, the Seedor Index [57] was calculated. The tibias from the birds euthanized at 21 days of age were used to determine dry matter in an oven with air circulation (105 °C). After this process, they were calcined for 8 h in a muffle furnace at 600 °C to obtain mineral matter (ash). The calcium and phosphorus contents in the bone were determined using the methodology described by [58]. Calcium concentration was measured by flame atomic absorption spectroscopy (FAAS), and phosphorus (P) was measured using ultraviolet–visible (UV-VIS) spectroscopy.

The left tibia was used to determine bone strength using the Brookfield CT3 texture analyzer. This equipment features a base that supports the epiphyseal areas of the bone, applying a force of 5 mm/s with a load of 200 kgf to the central region of the bone (diaphysis). During the strength determination, the tibia was placed on support in the same position each time. The results were expressed in kilogram-force (kgf) [59].

### 2.8. Gene Expression

At 21 days of age, one bird per experimental unit received stimulation through intraperitoneal administration of 1 mg of LPS (lipopolysaccharides from *E. coli*, Sigma-Aldrich, St. Louis, MO, USA) per live weight. After a 4 h period, the birds were sacrificed by cervical dislocation, and a fragment of the jejunum was immediately collected and immersed in a RNAlater™ stabilization solution (Invitrogen, Thermo Fisher Scientific, Carlsbad, CA, USA) and then stored in a freezer at −20 °C until RNA extraction. Total RNA was extracted using the QIAzol Lysis Reagent (Qiagen GmbH, Hilden, Germany). Approximately 70 mg of chicken intestine was weighed, crushed, and added to a microtube, free of DNase and RNase, containing 500 µL of Trizol. The samples were homogenized (vortex) and incubated at room temperature for 5 min. Next, 100 µL of chloroform was added, followed by manual homogenization for 15 s and incubation at room temperature for 3 min, followed by centrifugation for 15 min (12,000× *g* at 4 °C).

The aqueous phase was collected into a tube, and 250 µL of isopropanol was added, followed by a 10 min incubation (room temperature), manual homogenization, and centrifugation for 15 min at 12,000× *g* at 4 °C. The supernatant was discarded, and the precipitate was washed with 1 mL of 75% ethanol. The samples were centrifuged once again at 7500× *g* for 5 min, and the supernatant was discarded to dry the pellet for 15 min and then resuspended in DNase- and RNase-free ultrapure water and incubated at 60 °C for 15 min.

The RNA concentrations for the cytokines (IL-1β and IL-10), calbindin-D28K (CALB-D28K), and β-actin (endogenous control gene) were measured using a NanoDrop™ Lite spectrophotometer (Thermo Fisher Scientific, Waltham, MA, USA) at a wavelength of 260/280 nm. The integrity of the RNA was evaluated on 1% agarose gel stained with SYBR Safe™ DNA Gel Stain (Invitrogen, Thermo Fisher Scientific, Carlsbad, CA, USA) and visualized under ultraviolet transilluminator.

For cDNA synthesis, the QuantiTect Reverse Transcription Kit (Qiagen GmbH, Hilden, Germany) was used according to the manufacturer’s instructions, with 1 µg of RNA and a final volume of 20 µL. To remove potential genomic DNA residues, each sample was treated with 2 µL of gDNA Wipeout Buffer (Qiagen GmbH, Hilden, Germany) and incubated at 45 °C for 2 min. After removing genomic DNA, 4 µL of Quantiscript RT Buffer, 1 µL of RT Primer Mix, and 1 µL of Quantiscript Reverse Transcriptase were added. The reverse transcription reaction was incubated for 15 min at 42 °C, followed by 95 °C for 3 min, and immediately placed on ice. The samples were measured using the NanoDrop™ Lite spectrophotometer (Thermo Fisher Scientific, Carlsbad, CA, USA) and stored at −20 °C until use.

The primers/oligonucleotides used in the reactions were obtained from published studies on *Gallus gallus* (Table 3). The primer sequences were aligned using the BLAST (Basic Local Alignment Search Tool) algorithm in the NCBI database (http://www.ncbi.nlm.nih.gov/BLAST, accessed on 17 April 2024). The β-actin gene was considered a reference/housekeeping gene, and its stability across treatments was evaluated using the Statistica^®^ 7.0 program.

To evaluate the amplification efficiency of each primer, serial dilutions of the cDNA sample pool containing all treatments were performed using different concentrations of primers. For β-actin, CALB-D28K, and IL-10, 200 ng of cDNA and 400 nM of primer were used, and for IL-1β, 200 ng of cDNA and 600 nM of primer were used.

qRT-PCR analyses were conducted on a Rotor-Gene Q (Qiagen GmbH, Hilden, Germany) using the QuantiNova SYBR Green PCR Kit (Qiagen GmbH, Hilden, Germany) in duplicates. The total reaction volume was 20 µL. The amplification conditions were as follows: 95 °C for 2 min, followed by 40 cycles at 95 °C for 5 s, and 60 °C for 10 s. The melt curve of the reaction products was obtained to determine the specificity of the reactions.

Relative gene expression data were recorded as Ct (cycle threshold) values and normalized using the average Ct values obtained for the reference gene in each sample, for each treatment, and for each target gene, as recommended by [61]. The 2^−ΔCt^ method [60] was used for the relative quantification of gene expression (expressed as arbitrary units, AU).

### 2.9. Statistical Analyses

The data were evaluated for residual normality using the Shapiro–Wilk test and variance homogeneity using Levene’s test, both conducted through the Univariate procedure. For data with a normal distribution, a two-way analysis of variance (ANOVA) was performed to assess the effects of 1,25(OH)_2_D_3_-G supplementation in breeders and broilers, as well as the potential interactions between these factors. When significant effects were detected, treatment means for breeders were compared using the F-test, while treatment means for broilers were compared using the Student–Newman–Keuls test. All analyses were conducted using SAS for academics software (version 9.4 OnDemand), with a significance level of 5% [62].

## 3. Results

### 3.1. Performance

The performance variables of broilers at 21 days of age showed no significant interaction (*p* > 0.05) between the effects of 1,25(OH)_2_D_3_-G in breeders and the supplementation of 1,25(OH)_2_D_3_-G in broiler diets. However, breeders supplemented with 1,25(OH)_2_D_3_-G produced heavier broiler chicks (*p* = 0.001) at housing (Table 4). A significant effect of 1,25(OH)_2_D_3_-G inclusion in broiler diets was observed, resulting in a significant reduction in the feed conversion ratio (*p* = 0.001) at 21 days of age compared to broilers not receiving the supplementation.

### 3.2. Intestinal Histomorphometry

Villus height (*p* < 0.004) and absorption area (*p* = 0.001) in the jejunum of 21-day-old broilers showed a significant interaction between the effects of 1,25(OH)_2_D_3_-G supplementation in breeders and broiler diets (Table 5). When analyzing the interaction between 1,25(OH)_2_D_3_-G supplementation in breeder and broiler diets (Table 6), the inclusion of this additive in both diets was not sufficient to increase villus height and absorption area in broilers at 21 days of age. Broilers from breeders not supplemented with 1,25(OH)_2_D_3_-G had a greater villus height and absorption area when also not supplemented with the same additive.

### 3.3. Biochemical Variables

Calcium and phosphorus concentrations, as well as enzymatic variables, in 21-day-old broilers showed no significant interaction (*p* > 0.05) between the effects of 1,25(OH)_2_D_3_-G in breeders and the supplementation of 1,25(OH)_2_D_3_-G in broiler diets. However, 1,25(OH)_2_D_3_-G supplementation in the breeder diet resulted in higher alkaline phosphatase activity (*p* = 0.004) in broilers at 21 days of age (Table 7).

### 3.4. Bone Quality and Tibia Proximal Composition

No significant differences (*p* > 0.05) were identified for the Seedor index and tibia breaking strength in broilers at 21 days of age (Table 8). Calcium concentrations in broilers at 21 days showed a significant interaction (*p* = 0.006) between the effects of 1,25(OH)_2_D_3_-G in breeders and the supplementation of 1,25(OH)_2_D_3_-G in broiler diets. The inclusion of 100 mg/kg of 1,25(OH)_2_D_3_-G in broiler diets at 21 days resulted in lower calcium (*p* < 0.005) and phosphorus (*p* = 0.044) concentrations in the tibia (Table 9). Analyzing the interaction of 1,25(OH)_2_D_3_-G supplementation between breeders and broilers, it was observed that the inclusion of 100 mg/kg of 1,25(OH)_2_D_3_-G in both breeder and broiler diets resulted in lower calcium concentrations (*p* = 0.003) in the tibias of broilers at 21 days of age (Table 10).

### 3.5. Gene Expression

Gene expression of calbindin D28K and interleukins-10 and -1β in the jejunum of broilers at 21 days showed no significant interaction (*p* > 0.05) between the effects of 1,25(OH)_2_D_3_-G in breeders and 1,25(OH)_2_D_3_-G supplementation in broiler diets. However, higher expressions of calbindin D28K (*p* < 0.001), interleukin-10 (*p* = 0.001), and interleukin-1β (*p* = 0.001) were observed in the jejunum of broilers from breeders supplemented with 1,25(OH)_2_D_3_-G. Additionally, the inclusion of 1,25(OH)_2_D_3_-G in broiler diets (*p* = 0.047) indicated increased expression of calbindin D28K in the jejunum of birds receiving 50 and 100 mg/kg of 1,25(OH)_2_D_3_-G in their diets (Table 11).

## 4. Discussion

The results of this study indicate that the supplementation of breeders with 1,25(OH)_2_D_3_-G contributed to bone development and increased chick weight at hatch, highlighting the crucial role of this metabolite in nutrient transfer to the embryo [63]. By consuming a diet enriched with 1,25(OH)_2_D_3_-G, the breeder more efficiently absorbed calcium and phosphorus in the intestine, increasing their concentration in the bloodstream [64,65]. Since breeders primarily rely on dietary calcium for eggshell formation, without intense mobilization of medullary bone, 1,25(OH)_2_D_3_-G supplementation was essential to ensuring an adequate supply of minerals for both the eggshell and the embryo [66,67].

With a greater supply of calcium and phosphorus in the egg, the shell maintained its strength, protecting the embryo, while the yolk, enriched with these nutrients and 1,25(OH)_2_D_3_-G, provided support for chick bone development [64,65,68]. During incubation, the embryo first absorbed minerals from the yolk and later mobilized calcium from the eggshell for bone mineralization [68,69]. The 1,25(OH)_2_D_3_-G present in the egg stimulated the expression of calcium transporters in the intestinal epithelium of the embryo, enhancing calcium absorption [64,65,69]. Additionally, it promoted osteoblast activation, facilitating bone matrix deposition and resulting in stronger bones at hatch [70,71].

Although there are no studies directly exploring the use of 1,25-Dihydroxycholecalciferol (1,25(OH)_2_D_3_ in the diet of broiler breeders, related research suggests that the supplementation of vitamin D_3_ metabolites can improve egg quality and chick development [63,64,65]. The inclusion of 25-hydroxycholecalciferol (25(OH)D_3_), a direct precursor of 1,25(OH)_2_D_3_, in breeder diets enhances calcium and phosphorus absorption, resulting in eggs with stronger shells and higher mineral content [68,69]. Consequently, embryos benefit from an improved supply of these nutrients, which may lead to chicks with better bone health [70,71]. Additionally, supplementation with 25(OH)D_3_ has been shown to reduce the incidence of skeletal abnormalities in chicks, indicating a positive impact on early bone development [63,64]. These findings suggest that adding active vitamin D_3_ metabolites to breeder diets is an effective strategy to enhance egg quality and support the healthy growth of chicks [63,71].

The improved feed conversion in broiler chickens at 21 days of age with supplementation of 50 and 100 mg/kg of 1,25(OH)_2_D_3_-G suggests that feed conversion may be related to mechanisms distinct from intestinal morphology, such as the regulation of calcium and phosphorus metabolism, greater enzymatic efficiency, or other physiological processes [32]. This contradiction reinforces the complexity of the interaction between nutritional supplementation and physiological parameters, highlighting the need for further investigation into the underlying mechanisms [1,27]. Similarly, [3] observed improved feed conversion when broilers were supplemented with 1 and 2 µg of 1,25(OH)_2_D_3_-G from 1 to 21 days of age. These findings suggest that the addition of 1,25(OH)_2_D_3_-G can lead to significant improvements in performance during the growth phase, aligning with the results of the present study.

Although the supplementation of 1,25(OH)_2_D_3_-G did not increase intestinal villus height or absorption area, the observed improvements in feed conversion suggest that the additive may have enhanced the efficiency of metabolic processes involved in nutrient absorption and utilization [4,5,9,15]. Thus, despite the absence of structural changes in the intestine, supplementation appears to have optimized nutrient utilization, which is reflected in the improved feed conversion of broilers [5,20]

Moreover, supplementation may have influenced other physiological aspects, such as immunomodulation and maintenance of intestinal mucosal integrity, indirectly contributing to the efficiency of digestion and nutrient absorption [10]. These factors, combined, might have allowed the broilers to better utilize the nutrients available in the diet, resulting in improved feed conversion, even without a direct increase in villus height or absorption area [32]. Therefore, the improvement in feed conversion may be more closely associated with overall metabolic efficiency and optimization of the absorption of essential nutrients like calcium and phosphorus, rather than visible structural changes in the intestine [4,15,19,20]

The reduction in alkaline phosphatase activity in 21-day-old broiler chickens from breeders supplemented with 1,25(OH)_2_D_3_-G remained within the range considered normal for this age, suggesting a physiological adjustment associated with this supplementation protocol. Alkaline phosphatase is an essential enzyme for bone metabolism, and its activity generally reflects the rate of bone remodeling and mineralization [30,72]. Lower activity of this enzyme may indicate a reduced demand for phosphate mobilization to the bone, which could be related to a dynamic balance between mineral absorption and utilization [73]. Additionally, the regulation of alkaline phosphatase may be influenced by hormonal mechanisms mediated by 1,25(OH)_2_D_3_, which modulates the expression of genes involved in bone homeostasis [27]. These findings suggest that supplementation may have adjusted enzymatic activity according to the metabolic needs of the birds during this early growth phase [1,30].

The absence of changes in tibia bone strength in broilers at 21 days of age suggests that, despite the observed changes in alkaline phosphatase activity and calcium and phosphorus deposition in response to 1,25(OH)_2_D_3_-G supplementation, these changes were not reflected in detectable differences in bone strength [15,26,74]. The lack of distinction in bone strength between the groups may be attributed to various factors, including the complexity of bone and mineral metabolism, the influence of other dietary components, and genetic factors [30,72,75]. Moreover, it is important to consider that bone strength is influenced by various factors, such as bone quality and density, trabecular and cortical structure, as well as mineral composition [15,32,76]. Subtle changes in some of these aspects may not be detected through direct measurements of mineral deposition or enzymatic activity [75].

The reduction in calcium and phosphorus retention in the tibia with the higher dose of 1,25(OH)_2_D_3_-G suggests a possible activation of regulatory mechanisms that favored bone resorption over deposition [27]. At higher concentrations, 1,25(OH)_2_D_3_ may have stimulated RANKL expression, increasing osteoclastic activity and promoting the mobilization of these minerals to maintain systemic balance [21,30]. This process may have reduced mineral retention in the tibia without improving bone strength, indicating that supplementation influences bone metabolism dynamics in a complex manner, depending on the dose used [27,30,34].

Supplementation of 1,25(OH)_2_D_3_-G in the breeder diet had significant impacts on gene expression in broilers, suggesting a direct influence on physiological processes related to immune response and mineral metabolism [5,10,17,46]. The increase in calbindin D28K expression in broilers at 21 days of age, especially when supplemented with 50 mg/kg of 1,25(OH)_2_D_3_-G, suggested modulation of calcium transport in the intestine in response to the supplementation [77,78]. This means that there was increased production of this protein in intestinal cells and possibly a greater contribution to the bone development of growing broiler chickens [21,34,79].

Additionally, it is interesting to highlight that the calbindin D28K gene is associated with neuronal function, playing an essential role in the development and maintenance of brain health in birds [80]. When 1,25(OH)_2_D_3_-G binds to the VDR receptor, there is an increase in the expression of the calbindin D28K gene [79,81,82,83]. This results in increased production of the calbindin-D28K protein, which plays multiple roles in the nervous system, including regulating intracellular calcium and modulating neurotransmission [5,19,20]. This increase in the production of the calbindin-D28K protein can, therefore, significantly improve brain function in birds, contributing to better cognitive performance and healthier neurological development over time [77].

In this context, healthier neurological development is associated with more efficient and rapid growth in birds [32]. Optimized brain function can enhance nutrient absorption capacity, driving more vigorous growth and more efficient feed conversion [17,46]. Additionally, birds with improved brain function tend to exhibit a lower incidence of abnormal or stressful behaviors, which contributes to the overall well-being of the birds [39]. Additionally, a robust nervous system can strengthen the immune system of birds, making them more resistant to diseases and infections [82]. This, in turn, can reduce the need for antibiotics and medications, promoting a more sustainable and healthier production [10].

## 5. Conclusions

Supplementation with 100 mg/kg of 1,25(OH)_2_D_3_-G in breeder diets resulted in heavier broiler chicks at hatch, indicating improved nutrient transfer from the breeders to the eggs. Additionally, these chicks showed higher levels of interleukin 1β, interleukin 10, and calbindin D28K at 21 days of age, suggesting enhanced immune response and intestinal health. These health benefits extend into the growth phase, as broilers supplemented with 50 and 100 mg/kg of 1,25(OH)_2_D_3_-G exhibited better feed conversion ratios, reflecting more efficient nutrient utilization. This indicates that the supplementation not only improves the initial health of the chicks but also supports more efficient and healthy growth during the later stages.

## Figures and Tables

**Table 1 vetsci-12-00434-t001:** Ingredient composition and calculated nutrient content of experimental diets.

Ingredients, g/kg	1 to 7 Day	8 to 21 Day
Corn (7.88%)	504.76	525.03
Soybean Meal (46%)	423.88	402.13
Degummed Soybean oil	29.70	33.66
Dicalcium Phosphate	17.86	16.37
Limestone	9.39	8.64
Salt	4.02	3.68
Sodium Bicarbonate	1.00	1.50
Lysine Sulphate (60%)	1.70	1.75
DL-Methionine (99%)	3.27	3.09
L-Threonine (99%)	0.47	0.42
Choline Chloride (60%)	0.50	0.50
Adsorbent ^1^	1.00	1.00
Vitamin Premix ^2^	1.30	1.00
Mineral Premix ^3^	0.50	0.50
Anticoccidial ^4^	0.55	0.55
Enramycin ^5^		0.08
Inert (kaolin) ^6^	0.10	0.10
Calculated Nutrient Composition, g/kg		
Metabolisable Energy, MJ/kg	12.552	12.761
Crude Protein	240.471	231.848
Digestible Lysine	13.000	12.50
dMet+Cys ^7^	9.620	9.250
Digestible Threonine	8.580	8.250
Digestible Valine	10.000	9.630
Digestible Tryptophan	2.804	2.688
Digestible Arginine	15.196	14.573
Calcium	9.500	8.780
Available Phosphorus	4.500	4.190
Sodium	2.000	2.000
Potassium	9.372	9.039

^1^ Adsorbent: bentonite. ^2^ Vitamin supplement: composition per kg of product: vitamin A (min)—4.29 mg; vitamin D_3_ (min)—0.13 mg; vitamin E (min)—47.91 mg; vitamin K_3_ (min)—3.90 mg; vitamin B_1_ (min)—2.99 mg; vitamin B2 (min)—9.10 mg; pantothenic acid (min)—15.60 mg; vitamin B_6_ (min)—5.20 mg; vitamin B_12_ (min)—32.50 mg; niacin (min)—78.00 mg; folic acid (min)—2.60 mg; biotin (min)—0.33 mg; selenium (min)—0.39 mg. Vitamin supplement: composition per kg of diet initially: growth and finishing feeds: vitamin A—3.3 mg; vitamin D_3_—40.1 mg, 36.85 mg, 36.85 mg, and 36.85 mg; vitamin E—36.85 mg; vitamin K_3_—3.00 mg; thiamine (B_1_)—2.30 mg; riboflavin (B_2_)—7.00 mg; pyridoxine (B_6_)—4.00 mg; cyanocobalamin (B_12_)—25.00 mg; pantothenic acid (B_5_)—12.00 mg; niacin (B_3_)—60.00 mg; folic acid (B_9_)—2.00 mg; biotin (B_7_)—0.25 mg; selenium—0.30 g. ^3^ Mineral supplement: composition per kg of product: iron (min)—50 g; copper (min)—10 g; manganese (min)—65 g; zinc (min)—65 g; iodine (min)—1.000 mg. ^4^ Anticoccidial was as follows: From 1 to 21 days of age, Salinomycin 12% (Coxistac 12%) was used. From 22 to 35 days of age, Salinomycin 24% (Salinocox 24%) was used. ^5^ Enramycin 8% (Enradin 8%). ^6^ Inert based on kaolin, with the inclusion of 1,25(OH)_2_D_3_-G as a weight-for-weight substitution by the inert. ^7^ dMet+Cys: digestible methionine + cysteine.

**Table 2 vetsci-12-00434-t002:** Supplementation of vitamin D_3_ (1,25(OH)_2_D_3_-G) in broilers diets from breeders supplemented with the same additive.

Vitamin D_3_ Breeders, mg/kg	Vitamin D_3_ Broilers, mg/kg
0.0	0.0
0.0	50
0.0	100
100	0.0
100	50
100	100

**Table 3 vetsci-12-00434-t003:** Genes and primer sequences for gene expression analysis using qPCR.

Gene	Primer Sequence (5′→3′)	Amplicon Size (bp)	Reference
β-actina	F: TTCTTTTGGCGCTTGACTCA R: GCGTTCGCTCCAACATGTT	88	[60]
CALB-D28K	F: TTGGCACTGAAATCCCACTGAA R: CATGCCAAGACCAAGGCTGA	116	NM_205513.2
IL-1β	F: GCTCTACATGTCGTGTGTGATGAG R: TGTCGATGTCCCGCATGA	80	NM_204524.2
IL-10	F: CATGCTGCTGGGCCTGAA R: CGTCTCCTTGATCTGCTTGATG	94	NM_001004414.4

CALB-D28K: calbindin D28K; IL-1β: interleukin-1 beta; IL-10: interleukin 10.

**Table 4 vetsci-12-00434-t004:** Performance of broilers at 21 days of age fed diets supplemented or not with vitamin D_3_ (1,25(OH)_2_D_3_-G) from breeders supplemented or not with the same additive.

Vitamin D_3_ Levels, mg/kg	Hatching BW, g/Bird	FI, g/Bird	BWG, g/Bird	FCR, g FI/g BWG
Vitamin D_3_ Breeders				
0.0	47.73 ^b^	1154	917	1.261
100	48.48 ^a^	1157	923	1.254
SEM ^1^	0.03	5.93	4.31	0.01
Vitamin D_3_ Broilers				
0.0	48.12	1167	914	1.279 ^B^
50	48.14	1152	920	1.252 ^A^
100	48.06	1149	925	1.242 ^A^
SEM	0.09	5.89	4.33	0.01
*p*-Value				
Vitamin D_3_ Breeders	0.001	0.812	0.480	0.351
Vitamin D_3_ Broilers	0.893	0.443	0.603	0.001
Vitamin D_3_ Breeders × Vitamin D_3_ Broilers Interaction	0.737	0.672	0.845	0.245

^ab^: means followed by different lowercase letters in the column differ by F test at 5% significance. ^AB^: means followed by different uppercase letters in the column differ by Student–Newman–Keuls test at 5% significance. ^1^ SEM: standard error of the mean. FI: feed intake; BW: body weight; BWG: body weight gain; FCR: feed.

**Table 5 vetsci-12-00434-t005:** Histomorphometry of the jejunum of broilers at 21 days of age fed diets supplemented or not with vitamin D_3_ (1,25(OH)_2_D_3_-G) from breeders supplemented or not with the same additive.

Vitamin D_3_ Levels, mg/kg	VH, µm	CD, µm	AA, µm^2^	VH/CD, µm
Vitamin D_3_ Breeders				
0.0	780.50	49.17	16.43	15.89
100	745.51	44.51	16.17	17.52
SEM ^1^	17.78	1.53	0.53	0.60
Vitamin D_3_ Broilers				
0.0	804.82	46.42	17.44	17.14
50	761.37	49.58	15.55	15.90
100	724.36	44.17	15.92	17.09
SEM	17.48	1.54	0.52	0.61
*p*-Value				
Vitamin D_3_ Breeders	0.205	0.095	0.766	0.181
Vitamin D_3_ Broilers	0.089	0.315	0.329	0.304
Vitamin D_3_ Breeders × Vitamin D_3_ Broilers Interaction	0.004	0.105	0.001	0.810

^1^ SEM: standard error of the mean. Vit: vitamin. VH: villus height; CD: crypt depth, AA: absorption area; VH/CD: villus-height-to-crypt-depth ratio.

**Table 6 vetsci-12-00434-t006:** Breakdown of the interaction between vitamin D_3_ (1,25(OH)_2_D_3_-G) supplementation in the diets of breeders and broilers and its impact on villus height and absorption area in the jejunum of 21-day-old broilers.

	Villus Height, µm				Absorption Area, µm^2^			
Vitamin D_3_ Broilers, mg/kg		Vitamin D_3_ Breeders, mg/kg						
	0.0	100	SEM	*p*-Value	0.0	100	SEM	*p*-Value
0.0	898.52 ^aA^	722.84 ^B^	18.31	0.004	19.74 ^aA^	15.13 ^B^	0.72	0.004
50	780.15 ^ab^	742.60	36.14	0.599	15.76 ^b^	15.34	0.94	0.823
100	677.65 ^b^	771.08	24.21	0.066	13.80 ^bB^	18.03 ^A^	0.71	0.008
SEM ^1^	20.10	20.03			0.48	0.58		
*p*-Value	0.002	0.660			0.001	0.180		

^ab^: means followed by different lowercase letters in the columns differ by the Student–Newman–Keuls test at 5%. ^AB^: means followed by uppercase letters in the row differ by the F-test at 5%. ^1^ SEM: standard error of the mean.

**Table 7 vetsci-12-00434-t007:** Calcium, phosphorus concentrations, and serum enzyme activity of broilers at 21 days of age fed diets supplemented or not with vitamin D_3_ (1,25(OH)_2_D_3_-G) from breeders supplemented or not with the same additive.

Vitamin D_3_ Levels, mg/kg	Ca mg/dL	P mg/dL	LDH U/L	CPK U/L	ALP U/L
Vitamin D_3_ Breeders					
0.0	9.48	7.50	683	2162	26,500 ^a^
100	9.09	7.16	642	2872	20,920 ^b^
SEM ^1^	0.11	0.13	30.56	123.54	412.70
Vitamin D_3_ Broilers					
0.0	9.16	7.24	662	2349	23,561
50	9.70	7.48	671	3233	24,057
100	9.01	7.27	654	1904	23,684
SEM	0.12	0.13	31.05	151.28	316.01
*p*-Value					
Vitamin D_3_ Breeders	0.411	0.184	0.505	0.320	0.004
Vitamin D_3_ Broilers	0.492	0.691	0.965	0.274	0.962
Vitamin D_3_ Breeders × Vitamin D_3_ Broilers Interaction	0.758	0.545	0.367	0.611	0.529

^ab^: means followed by lowercase letters in the column differ according to the F-test at 5% significance. ^1^ SEM: standard error of the mean. Ca: calcium; P: phosphorus. LDH: lactate dehydrogenase; CPK: creatine phosphokinase; ALP: alkaline phosphatase.

**Table 8 vetsci-12-00434-t008:** Seedor index and tibia breaking strength of broilers at 21 days of age fed diets supplemented or not with vitamin D_3_ (1,25(OH)_2_D_3_-G) from breeders supplemented or not with the same additive.

Vitamin D_3_ Levels, mg/kg	Seedor Index	Breaking Strength, kg
Vitamin D_3_ Breeders		
0.0	68.82	22.45
100	70.18	22.28
SEM ^1^	1.02	0.53
Vitamin D_3_ Broilers		
0.0	68.43	21.89
50	67.26	21.41
100	72.78	23.77
SEM	0.97	0.52
*p*-Value		
Vitamin D_3_ Breeders	0.453	0.879
Vitamin D_3_ Broilers	0.057	0.142
Vitamin D_3_ Breeders × Vitamin D_3_ Broilers Interaction	0.505	0.726

^1^ SEM: standard error of the mean.

**Table 9 vetsci-12-00434-t009:** Bromatological composition of the tibia expressed in dry matter of broilers at 21 days of age fed diets supplemented or not with vitamin D_3_ (1,25(OH)_2_D_3_-G) from breeders supplemented or not with the same additive.

Vitamin D_3_ Levels, mg/kg	Dry Matter, %	Phosphorus, %	Calcium, %
Vitamin D_3_ Breeders			
0.0	40.12	11.34	16.23
100	40.70	11.31	16.27
SEM ^1^	0.33	0.29	0.36
Vitamin D_3_ Broilers			
0.0	39.72	11.68 ^A^	17.51 ^A^
50	40.71	11.96 ^A^	16.64 ^A^
100	40.83	10.33 ^B^	14.59 ^B^
SEM	0.33	0.27	0.31
*p*-Value			
Vitamin D_3_ Breeders	0.344	0.966	0.951
Vitamin D_3_ Broilers	0.306	0.044	0.005
Vitamin D_3_ Breeders × Vitamin D_3_ Broilers Interaction	0.487	0.900	0.006

^AB^: means followed by different uppercase letters in the column differ by Student–Newman–Keuls test at 5% significance. ^1^ SEM: standard error of the mean.

**Table 10 vetsci-12-00434-t010:** Breakdown of the interaction between vitamin D_3_ (1,25(OH)_2_D_3_-G) supplementation in the diets of breeders and broilers and its impact on calcium concentrations in 21-day-old broilers.

Vitamin D_3_ Broilers, mg/kg	Vitamin D_3_ Breeders, mg/kg			
	0.0	100	SEM	*p*-Value
0.0	16.78	18.25 ^a^	0.45	0.118
50	15.98	17.31 ^a^	0.63	0.292
100	15.94 ^A^	13.25 ^bB^	0.42	0.005
SEM ^1^	0.37	0.45		
*p*-Value	0.567	0.003		

^ab^: means followed by different lowercase letters in the column differ from each other according to the F-test at 5% significance. ^AB^: means followed by uppercase letters in the column differ from each other according to the Student–Newman–Keuls test at 5% significance. ^1^ SEM: standard error of the mean.

**Table 11 vetsci-12-00434-t011:** Jejunum gene expression in broilers at 21 days of age fed diets supplemented or not with vitamin D_3_ (1,25(OH)_2_D_3_-G) from breeders supplemented or not with the same additive.

Vitamin D_3_ Levels, mg/kg	Calbindin D28K	Interleukin 10	Interleukin 1β
Vitamin D_3_ Breeders			
0.0	0.239 ^b^	1.557 ^b^	1.030 ^b^
100	0.358 ^a^	1.820 ^a^	1.366 ^a^
SEM ^1^	0.02	0.12	0.07
Vitamin D_3_ Broilers			
0.0	0.273 ^B^	1.781	1.255
50	0.318 ^A^	1.707	1.108
100	0.306 ^A^	1.599	1.207
SEM	0.02	0.13	0.07
*p*-Value			
Vitamin D_3_ Breeders	0.001	0.001	0.003
Vitamin D_3_ Broilers	0.047	0.176	0.461
Vitamin D_3_ Breeders × Vitamin D_3_ Broilers Interaction	0.122	0.227	0.897

^ab^: means followed by different lowercase letters in the column differ from each other according to the F-test at 5% significance. ^AB^: means followed by uppercase letters in the column differ from each other according to the Student–Newman–Keuls test at 5% significance. ^1^ SEM: standard error of the mean.

## Data Availability

Data will be made available on reasonable request.

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
