# Peer review of "Effects of 1,25-Dihydroxycholecalciferol Glycoside Supplementation on the Growth, Intestinal Health, and Immunity of Broilers from Breeders Supplemented or Not with the Same Additive"

_vetsci, 2025, doi:10.3390/vetsci12050434_

Round 1

Reviewer 1 Report

Comments and Suggestions for Authors

The manuscript is suitable for publication. Please check the data in the footer of the ration table.

Author Response

Dear Reviewer,

Thank you for your valuable contribution. The adjustments have been made as suggested.

Best regards,

Andrade et al

Reviewer 2 Report

Comments and Suggestions for Authors

The study explores the use of plant bioactive-based Vit D3-G as feed additives, an alternative and complementary solution for producers.  The use of a controlled experimental design with defined treatments and replication is commendable, ensuring reliable data collection. The results are presented systematically, highlighting findings for both productivity and intestinal structure with immunity in chickens.
Suggestions for Improvement:
Background Clarity: The introduction section is lacking with the practices in the feed industry regarding basic principles of dietary supplementation with vitamin D3 in the form of 25-or 1.25 dihydroxycholecalciferol.  Considering the current standards in broiler feed production which have proved great success over a half century, what was the basis for further addition of another vitamin D3 source derived from phytochemicals.   Is there a problem with level and type of the vitamin D3 in the feeding of commercial broilers? What are the related recommendations by scientific communities (book values) and the main broiler integrators producing over hundred millions of broiler chickens per annum? Addressing of these issues in the introduction would also provide new insight while easing the reader to grasp your purpose and comments that you made.

The relationship between maternal feeding and feeding of their progeny chicks with regard to vitamin D3 supplementation is lacking. Although the subject is original, researchers have not made satisfactory explanations. The fact that the offspring of the breeders given additional Vit D3-G are 1 g heavier than the unsupplemented ones at hatching addressed the importance of the subject.

Expand on why supplemental Vit D3-G in addition to conventional vit D3 preparations routinely used.  It seems that this further addition of Vit D3-G is expected to improve performance and mineral utilization in chickens. Is Vit D3-G is known to influence related physiological or metabolic responses to standard Vit D3 supplementation by vitamin premixes.

 Authors generally overestimate and reports in favour of Vit D3-G supplementation although it conveyed benefits only on FCR. However, the detrimental effects Vit D3-G supplementation on VH and AA in intestines contradicts with FCR findings, which warrants further clarification?

The significant reduction in tibia P and Ca retention with Vit D3-G supplementation is noticeable, but authors overlooks this issue. It would be useful to briefly explain the biological implications of these significant results.
            Although blood ALP level was reduced by Vit D3-G supplementation in broiler breeders after 40 weeks implementation, was this reduction within a healthy range for breeder hens, or could it indicate potential drawbacks over extended use or any sign of adverse effect?     The discussion on the negative impacts of Vit D3-G supplementation on bone mineralization and intestinal absorbtion surface area could be expanded. Why might Vit D3-G reduce these parameters?
Conclusions:       While the conclusion is supported by the data, it might benefit from a more balanced discussion. For instance, acknowledging the adverse effects of Vit D3-G  on bone mineralization and absorption surface reduction would provide a nuanced perspective. Enhanced immune response alone can not be the driver of improvement in FCR by 1% alone.

Comments on the Quality of English Language

quality of the English Language can be improved 

Author Response

Dear Reviewer,

Your suggestions differed significantly from those of the other reviewers, which made the adjustments somewhat challenging. However, we carefully considered all your comments and attempted to align them, where possible, with the recommendations of the other reviewers. If there are any essential points that still require revision, we are open to making further adjustments.

Best regards,
Andrade et al

Reviewer 3 Report

Comments and Suggestions for Authors

This kind of research study is important to poultry producers. This manuscript has only minor comments, and if the authors address them, it can be accepted for publication. 

  • line 2: The title is not attractive; please rewrite it
  • line 18: The abstract lacks some results to give the reader an excellent idea about the research fully and obtained results, please add some more data 
  • line 30-31: giving information about the interaction is not enough, please provide more information about the differences between the treatments 
  • Line 35: In the introduction, the style of numbering the references is not correct, as per the journal's style. 
  • Overall, the introduction is well-detailed and well-written, but please add the hypothesis of the study at the end of the introduction.
  • line 118: why the authors did choose this level of supplementation?
  • Lines 220-227 indicate the sensitivity and specificity of the spectrophotometric methods used, as well as the inter- and intra-coefficients of variation of the different procedures. 
  • The rest of the manuscript is fine and complete  

Author Response

(The authors gave the same response as above.)

Reviewer 4 Report

Comments and Suggestions for Authors

This manuscript provides important informations regarding 1,25-Dihydroxycholecalciferol glycoside and its effects on the performance and immunity of broilers, after minor corrections, I suggest to publish the article.

The Introduction is very well-written, congratulations!

Please add a picure of the measurement of the villus height/crypt depth etc. after line 206

Based on the gene expression results, can you tell weather the supplement up/down regulated the examined genes? (-/+ fold change values). If there is a possibility for that, please add this result. I also miss graphs especially for gene expression results. Try to put the existing results to a bar graph, and the up/down regulated part on a volcano plot, or something similar.

Author Response

Revisor, não foi possível incluir uma figura com a medida da altura das vilosidades e da profundidade das criptas, pois essas análises foram realizadas em outro laboratório. Da mesma forma, os resultados da expressão gênica também foram obtidos em um laboratório diferente, impossibilitando a inclusão dos valores de mudança de dobra (+/-). Entendemos a sugestão, mas acreditamos que a ausência dessas informações não compromete a validade do artigo, pois atende ao seu objetivo e apresenta os resultados com clareza. No entanto, se essas informações forem essenciais para a publicação, estamos abertos a discutir alternativas. Nossa abordagem é baseada em estudos sobre vitamina D₃, que normalmente apresentam esses resultados em formato de tabela. Agradecemos seu feedback, que será valioso para incorporar tais análises em estudos futuros conduzidos por nosso laboratório.

Round 2

Reviewer 2 Report

Comments and Suggestions for Authors

Thanks a lot to the authors for the alterations and clarifications that have been made to the paper according to many of my comments. The  authors' response and interpretations coincide with my original points  previously addressed. To conclude, the present format of the manuscript can be accepted for the publication.